# Segmenting white matter hyperintensities on isotropic three-dimensional Fluid Attenuated Inversion Recovery magnetic resonance images: Assessing deep learning tools on a Norwegian imaging database

**Martin Soria Røvang**[1,2]*, **Per Selnes**[3,4], **Bradley J. MacIntosh**[2,5,6], **Inge Rasmus Groote**[2,7], **Lene Pålhaugen**[3,4], **Carole Sudre**[8,9,10], **Tormod Fladby**[3,4], **Atle Bjørnerud**[2,8,11]

1 Division of Medicine and Laboratory Sciences, University of Oslo, Oslo, Norway, 2 Division of Radiology and Nuclear Medicine, Computational Radiology & Artificial Intelligence (CRAI), Oslo University Hospital, Oslo, Norway, 3 Department of Neurology, Akershus University Hospital, Lørenskog, Norway, 4 Institute of Clinical Medicine, Campus Ahus, University of Oslo, Oslo, Norway, 5 Department of Medical Biophysics, University of Toronto, Toronto, Canada, 6 Sandra Black Centre for Brain Resilience & Recovery, Hurvitz Brain Sciences Program, Physical Sciences Platform, Sunnybrook Research Institute, Toronto, Canada, 7 Department of Radiology, Vestfold Hospital Trust, Tønsberg, Norway, 8 Center for the Study of Human Cognition, Department of Psychology, University of Oslo, Oslo, Norway, 9 Centre for Medical Image Computing, University College London, London, United Kingdom, 10 School of Biomedical Engineering & Imaging Sciences, King's College London, London, United Kingdom, 11 Department of Physics, University of Oslo, Oslo, Norway

* roevma@ous-hf.no

**Data Availability Statement:** Data cannot be shared publicly because of ethics approval

## Abstract

An important step in the analysis of magnetic resonance imaging (MRI) data for neuroimaging is the automated segmentation of white matter hyperintensities (WMHs). Fluid Attenuated Inversion Recovery (FLAIR-weighted) is an MRI contrast that is particularly useful to visualize and quantify WMHs, a hallmark of cerebral small vessel disease and Alzheimer's disease (AD). In order to achieve high spatial resolution in each of the three voxel dimensions, clinical MRI protocols are evolving to a three-dimensional (3D) FLAIR-weighted acquisition. The current study details the deployment of deep learning tools to enable automated WMH segmentation and characterization from 3D FLAIR-weighted images acquired as part of a national AD imaging initiative. Based on data from the ongoing Norwegian Disease Dementia Initiation (DDI) multicenter study, two 3D models—one off-the-shelf from the NVIDIA nnU-Net framework and the other internally developed—were trained, validated, and tested. A third cutting-edge Deep Bayesian network model (HyperMapp3r) was implemented without any de-novo tuning to serve as a comparison architecture. The 2.5D in-house developed and 3D nnU-Net models were trained and validated in-house across five national collection sites among 441 participants from the DDI study, of whom 194 were men and whose average age was (64.91 +/- 9.32) years. Both an external dataset with 29 cases from a global collaborator and a held-out subset of the internal data from the 441 participants were used to test all three models. These test sets were evaluated independently. The ground truth human-in-the-loop segmentation was compared against five established WMH

restrictions not allowing data sharing with third parties outside those defined in patient informed consent. Model weights can be given upon request to the author: m.s.rovang@medisin.uio.no or Martin Sending, Head of Research Administration, Oslo University Hospital Responsible for establishing material transfer agreements at OUS marsen@ous-hf.no. Source code can be requested at the given e-mails but can also be found here: https://github.com/MartinRovang/WMH_segmentation.

**Funding:** MR,PS,IG,LP,CS,TF,AB Grant: EU Joint Programme - Neurodegenerative Disease Research (NFR 311993) https://www.neurodegenerationresearch.eu/ The funders had no role in study design, data collection and analysis, decision to publish, or preparation of the manuscript.

**Competing interests:** The authors have declared that no competing interests exist.

performance metrics. The 3D nnU-Net had the highest performance out of the three tested networks, outperforming both the internally developed 2.5D model and the SOTA Deep Bayesian network with an average dice similarity coefficient score of 0.76 +/- 0.16. Our findings demonstrate that WMH segmentation models can achieve high performance when trained exclusively on FLAIR input volumes that are 3D volumetric acquisitions. Single image input models are desirable for ease of deployment, as reflected in the current embedded clinical research project. The 3D nnU-Net had the highest performance, which suggests a way forward for our need to automate WMH segmentation while also evaluating performance metrics during on-going data collection and model retraining.

## Introduction

Alzheimer's disease (AD) is the most common type of dementia and is characterized by the deposition of neurotoxic amyloid plaques in the cerebral cortex [1]. Amyloid plaques can be identified using (amyloid) Positron Emission Tomography (PET) or by measuring the concentration of amyloid-β species in cerebrospinal fluid. Cerebrovascular small vessel disease (SVD) is an independent driver of dementia and the co-pathologies seen between AD and SVD [2]. Subcortical white matter hyperintensities (WMHs) of presumed vascular origin on T2-weighted MRI are an established surrogate marker for SVD [3], but WMHs are also considered a core feature of AD [4].

Quantifying WMH lesion load through segmentation on MRI is relevant to the characterization of AD and vascular cognitive impairment. However, this introduces a large workload in a neuroradiology service, and therefore, robust and automated segmentation software tools are desirable [5].

Deep learning has demonstrated clinical value across a wide range of imaging applications [6], including lesion segmentation in MRI. A recent report combined 2D T2-weighted fluid-attenuated inversion recovery (FLAIR-weighted) and 3D T1-weighted images to yield good WMH segmentation performance, arguably a state-of-the-art (SOTA) solution [7]. Other deep learning-based WMH segmentation publications trained the models using either fully or partly open-source datasets, for example, in the Medical Image Computing and Computer-assisted Intervention (MICCAI) WMH segmentation challenge [8] and model comparison study [9]. Open-source data ensures that results can be readily compared. However, it remains to be seen whether new data sources, such as those from other MRI systems and sites, will degrade performance. The MICCAI WMH segmentation challenge has dramatically increased research interest and collaboration, and there is a need to extend this work, i.e., build on the work that included 170 participants, five different scanners, three different vendors, and three hospitals in the Netherlands and Singapore. This previous effort relied on 3D T1-weighted and 2D multi-slice FLAIR image acquisitions.

There is broad interest in automated WMH detection and quantification tools. Legacy FLAIR data have been collected using 2D multi-slice sequences that yield high signal-to-noise ratio images at the expense of through-plane spatial resolution. Whereas, a 3D and isotropic voxel resolution FLAIR sequence is increasingly the norm. The current study aligns with an ongoing Norwegian nationwide effort aimed at the early detection of AD [10]. Due to the large number of participants, it is not feasible to manually segment WMH; this is compounded by the fact that a 3D FLAIR has roughly three times as many slices as a 2D FLAIR, which increases annotation time. Prior WMH research demonstrates the feasibility of segmentation based on

3D FLAIR images [11, 12]; there is a need to extend this research with fast and automated methods of which deep learning based approaches are highly relevant and enable large sample size cohorts. Therefore the primary objective of this work is to examine deep learning solutions that can yield robust segmentations across field strength and site.

The current study investigates three convolutional neural network (CNN) architectures that are applied to 3D-acquired FLAIR images. Two trained networks—one of which was created internally—were trained de-novo. Models were trained solely on 3D FLAIR-weighted data from the multicenter Norwegian cohort. Performance was compared against a recently published model, a Deep Bayesian network model (Hypermapp3r) [13], which has the distinction of requiring a FLAIR and a T1-weighted image when this model is run in inference mode. Training a neural network that uses both FLAIR and T1-weighted inputs offers some advantages, such as a higher signal-to-noise ratio and more information per participant. There are disadvantages too; notably, image alignment and ease of implementation. Furthermore, a single image input framework is conducive to large batches of MRI data. We hypothesized that the two in-house trained models that used 3D FLAIR-weighted images will achieve high WMH segmentation performance using widely accepted performance metrics for internal and external data sources.

## Materials and methods

In this section, we show the data statistics, information, and splits. We also show the methods used, such as pre-processing, networks, and prediction methods.

### Participants

The longitudinal Norwegian Disease Dementia Initiation (DDI) multicenter study, which enrolled 441 adult participants (194 men), had a mean age of (64.91 +/- 9.32) years at the time of enrollment. The DDI study includes five national sites with MRI performed on six different scanners from three different vendors. Details of the study population included are reported previously [14]. In short, individuals who reported cognitive concerns or as assessed by next of kin were recruited mainly by advertisement, from memory clinics, or from a previous study. Exclusion criteria were brain trauma or disorder, stroke, previous dementia diagnosis, severe psychiatric disease, a severe somatic disease that might influence cognitive functions, intellectual disability, or developmental disorders. Age-matched controls were recruited from advertisements or were patients admitted to the hospital for orthopedic surgery. We also included a second cohort of 29 adults that were scanned at a single site in the Czech Republic to serve as an external test set. All participants provided written consent, and the study has been approved by the regional ethics committee (REK SØ 2013/150).

### MRI data

3D FLAIR-weighted MR images were acquired at 1.5 T or 3 T. Relevant pulse sequence parameters are provided in Table 1. Table 2 provides the breakdown of data splitting. The intra-scanner mean intensity distribution is shown in Fig 1, which shows some differences between scanner types.

### WMH distribution

The median and the interquartile range (IQR) of the total WMH volume per participant were 2.85 mL (1.25–58.2) mL, with a total range of (0.11–78.26) mL. The distribution of total WMH in the internal test data had a median and IQR of 3.35 mL (1.14, 5.0) mL and a total range of

**Table 1. Dataset information for the internal dataset.**

| Institution | Model | Software | Field Strength | Repetition Time (msec) | Echo Time (msec) | Flip Angle (deg) | Resolution (x,y,z) (mm) |
|---|---|---|---|---|---|---|---|
| Scanner 1 | Avanto | syngo_MR_B19 | 1.5 | 1700 | 2.42 | 15 | (1.2,1.05,1.05) |
| Scanner 2 | Avanto | syngo_MR_B19 | 1.5 | 1700 | 2.42 | 15 | (1.2, 1.04, [1.0,1.04]) |
| Scanner 3 | Avanto | syngo_MR_B19 | 1.5 | 1700 | 2.42 | 15 | (1.2, 1.05, 1.05) |
| Scanner 4 | Avanto | syngo_MR_B19 | 1.5 | 1180 | 4.36 | 15 | (1.0, 1.0, 1.0) |
| Scanner 5 | Ingenia | [5.1.7,5.3.0.3] | 3.0 | [4.47, 4.91] | [2.218, 2.431] | 8 | ([1.0, 1.2], 1.2, 1) |
| Scanner 6 | Prisma | syngo_MR_E11 | 3.0 | 2200 | 1.47 | 8 | (1.0, 1.0, 1.0) |
| Scanner 7 | Skyra | syngo_MR_E11 | 3.0 | 2300.0 | 2.98 | 9 | (1.2, 1.0, 1.0) |
| Scanner 8 | Prisma | syngo_MR_E11 | 3.0 | 2200 | 1.47 | 8 | (1.0, 1.0, 1.0) |
| Scanner 8 | Ingenia | [5.1.7, 5.1.7.2] | 3.0 | [4.48,4.92] | [2.225, 2.45] | 8 | ([1.0, 1.2], 1.0, 1.0) |
| Scanner 9 | Skyra | syngo_MR_D113C | 3.0 | 2300 | 2.98 | 9 | (1.2, 1.0, 1.0) |
| Scanner 10 | Achieva | [3.2.1, 3.2.1.1] | 3.0 | [6.36, 6.72] | [3.02, 3.14] | 8 | ([1.0, 1.2], 1.0, [1.0, 1.01]) |
| Scanner 11 | Ingenia | [5.1.2, 5.1.7, 5.1.7.2] | 1.5 | [7.47, 7.90] | [3.40, 3.68] | 8 | (1.0, 1.0, 1.0) |
| Scanner 12 | Optima MR450w | [DV25.0_R01_1451.a. DV25.1_R03_1802.a] | 1.5 | [11.24, 11.38] | [5.0, 5.05] | 10 | (1.2, [1.0, 1.04], [1.0, 1.04]) |
| Scanner 13 | Skyra | [syngo_MR_D113C, syngo_MR_E11] | 3.0 | 2300 | [2.97, 2.98] | 9 | (1.2, [1.0, 1.01], [1.0, 1.01]) |
| External Scanner 1 | Prisma | syngo_MR_E11 | 3.0 | 2200 | 1.47 | 8 | (1.0, 1.0, 1.0) |

(0.15, 22.90) mL. The distribution for the total WMH in the external test data had a median and IQR of 0.986 mL (0.611, 2.652) mL and a total range of (0.123, 21.252) mL.

## Data annotation

Ground truth WMH annotations were generated as previously described. In brief, initial segmentation estimates were obtained using Gaussian mixture modeling [14, 15]. These automated segmentations were subsequently visually reviewed and approved by a domain expert neurologist (LP). For a subset of the cases, multiple iterations of the Gaussian mixture model were needed to exclude false positives, and a consensus review with a second neurologist (PS) was performed for cases with uncertain ground truth.

## Network architectures

The two prototype networks were 2.5D- and 3D nnU-Net models trained on the Norwegian cohort's 3D FLAIR-weighted data. The former is a model developed in-house and referred to as 2.5D to denote the role that neighboring slices/views contribute to the training and

**Table 2. Splitting of MRI cases for training, validation, and testing across the different MR systems.** The Avanto and Optima MR450 are 1.5 Tesla systems, and the remaining 3 Tesla systems.

| Data split | Skyra | Prisma | Ingenia | Optima MR450w | Achieva | Avanto |
|---|---|---|---|---|---|---|
| Training n = 300 | 28 | 11 | 155 | 62 | 31 | 13 |
| Validation n = 75 | 6 | 2 | 37 | 14 | 12 | 4 |
| Test (Internal) n = 66 | 9 | 4 | 27 | 14 | 6 | 6 |
| Test (External) n = 29 | - | 29 | - | - | - | - |

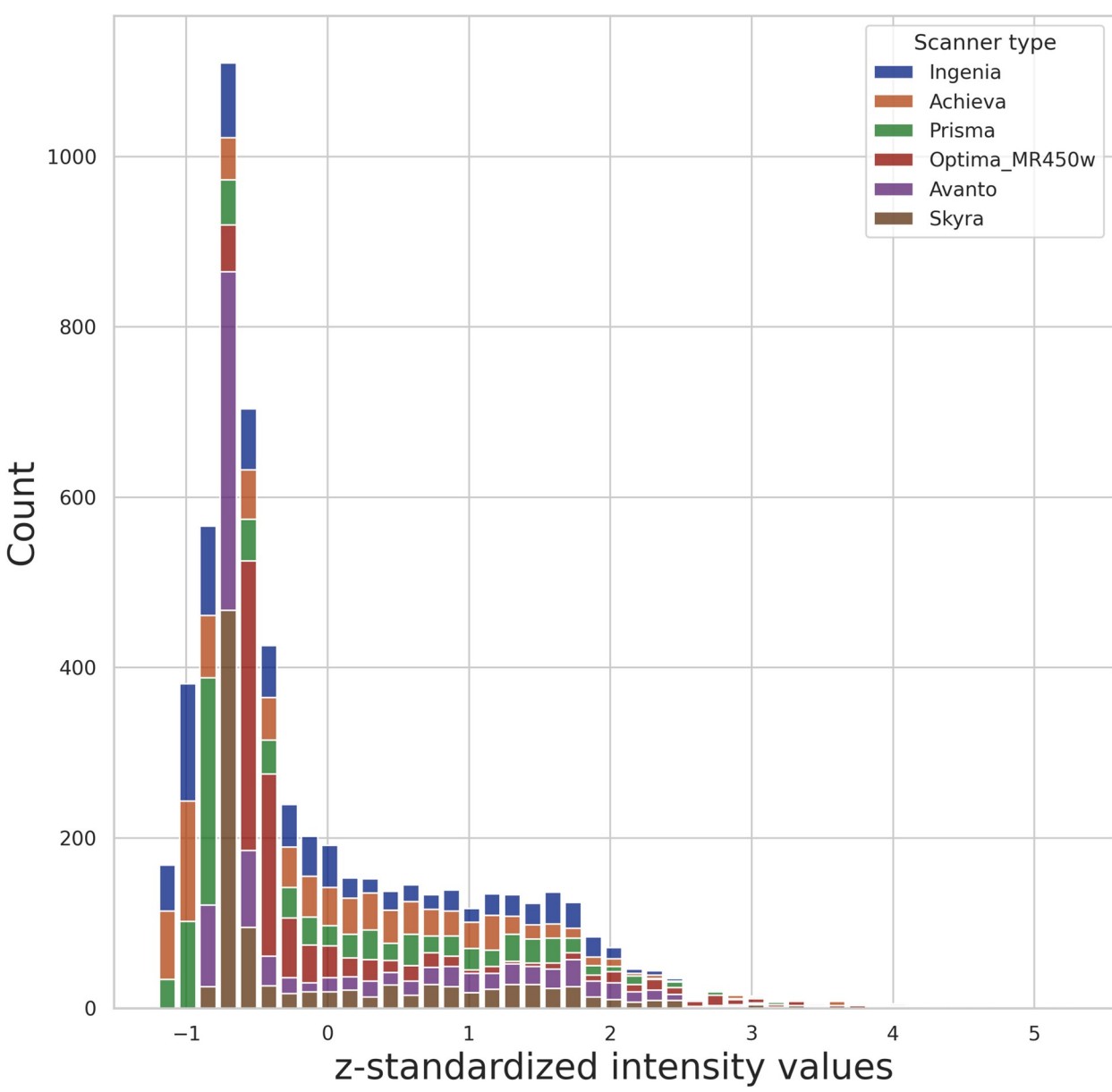

**Fig 1. Intensity distribution of the data.** The z-standardized intensity distributions for the foreground values across all participants for the different scanners included in the study for the internal training, validation, and test datasets.

inference [16]. The latter is the nnU-Net which has shown promising results for many different medical image segmentation problems [17]. A third model was also considered in this study; the Deep Bayesian networks (we will call this Deep Bayesian going forward), also called the Hypermapp3r model, which has recently been described and was considered a state-of-the-art WMH segmentation tool. Here, the Deep Bayesian was implemented "off the shelf" and was previously trained on two image inputs (i.e., 3D T1- and 2D FLAIR-weighted images), and no *de-novo* network training was performed for the current study.

**2.5D U-Net.** We used a 256x256 U-Net architecture [18] with encoding feature map sizes of *(32, 64, 256, 512)*. Each convolution block used the Mish activation function [19]. The upsampling in the decoding layers consisted of transposed convolutions [20]. Using a

2.5-dimensional U-Net configuration, three adjacent 3D FLAIR-weighted slices were used as input on separate channels, with a prediction being made on the center channel. This model had a total of 7.8 M parameter estimates and was developed in Pytorch [21]. Code can be found in [22].

**3D nnU-Net.** A 3D *DynUNet* from the Medical Open Network for Artificial Intelligence (MONAI) python library [23] was used as implemented in the 3D nnU-Net *NVIDIA NGC* catalog [24] V. 21.11.0, where "nn" stands for "no new" to denote a widely used and standardized U-Net implementation. We used default parameters for this model. The encoding feature map sizes were: (*32, 64, 128, 256, 320, 320)*. This model had a total of 31.2 M parameter estimates and was developed in Pytorch.

**Reference model: Deep Bayesian model (HyperMapp3r).** This comparison model is described as a Deep Bayesian using a 3D CNN based on the U-Net structure with Monte-Carlo(MC) dropout at each encoding layer; the dropout layers allow for confidence intervals to be estimates of uncertainty along with the predicted segmentation. The MC drop out a percentage of the layers for every inference making the model non-deterministic we are able to find a variance estimate of the prediction. The encoding feature map sizes were *(16, 32, 64, 128, 256)* and had a total of 515 K parameters (MC drop out). This model was developed in Keras [25, 26].

## Data pre-processing

Data was converted from Digital Imaging and Communications in Medicine (DICOM) to Neuroimaging Informatics Technology Initiative (NIfTI) files using Mcverter [27] and intensity bias-corrected using the N4 algorithm [28, 29]. Lesions below five voxels in diameter were considered to represent noise and were removed from the dataset [30] using the diameter_opening morphology function from scikit-image [31]. To reduce some of the noise, all intensity values in the FLAIR images that were less than zero were set to zero.

**2.5D U-Net.** Intensity values above zero were used for z-standardization. Then, 1/12 of the outermost slices from each plane were removed to decrease the amount of empty/non-informative slices used during training.

**3D nnU-Net.** The standard 3D nnU-Net preprocessing framework was used, which included z-normalization of intensity values above zero and resampling to an isotropic resolution of 1 mm$^3$.

**Deep Bayesian.** The pre-processing steps described in the Deep Bayesian model included N4 bias correction, skull-stripping for separating the brain from non-brain tissues [32], and z-normalization of the segmented brain volumes.

**Post-processing.** The models' probability outputs were thresholded at 0.5 to produce binary segmentation masks.

## Training & validation

In this section we summarize the main points of the training and validation for the different models used in our experiments.

**2.5D U-Net.** The loss and evaluation metrics were calculated by first finding the loss and scores in a mini-batch, which consisted of 8x3 individual slices selected from all three orthogonal planes of the MRI volumes from four randomly sampled scan volumes. We applied the same augmentation scheme as used in the 2017 MICCAI WMH Challenge-winning model [33]. The mini-batch array was flattened along the slice axis, and a slightly modified version of the Tversky focal loss function [34] was used, as specified in Eq 4.12 in [16].

Following the recommendations in [34], we used $\gamma = \frac{4}{3}$. In the same paper, $\alpha = 0.70$ and $\beta = 0.30$ was used with the aim of improving convergence by minimizing false negative predictions, but in our case we chose values $\alpha = 0.85$ and $\beta = 0.15$ to further minimize false negatives, based on pilot data in our previous work [16]. Finally, we set $\Omega = 1$ to avoid zero division in the absence of WMH.

During validation, the model weights were saved when the validation data had the highest $F_2$ score, defined by the generalized $F$ score given in Eq 4.5 in [16].

The parameter values for the $F$ score were set to $\Omega = 1$ to avoid division by zero and $\beta = 2$ (not to be confused with the Tversky $\beta$ parameter) to match the objective of the Tversky focal loss of optimizing for fewer false negatives.

After all the mini-batches for the epoch were completed, the result was set as the average over all the mini-batch step results. The $F_2$ scores were only calculated for the validation set. During the evaluation, the weights were saved at the maximum $F_2$. The model was trained for 43 epochs and then stopped manually since the validation metric was found to plateau and remain stable for >9 epochs.

The initial learning rate was $\eta = 0.0001$. A learning rate scheduler reduced the learning rate by a factor of 0.2 every five plateau epochs. The learning rate was reduced if the validation loss did not decrease for five epochs. A mini-batch size of 8x3 random slices was used, with eight slices taken from three orientations of the volume.

We use an Adam optimizer [35] with the following parameters: betas = (0.9,0.999) eps = $1e^{-8}$ and weight decay = 0. The MONAI function "set determinism" with seed 0 and "None" as additional settings, was used for reproducibility. All of these parameter values were selected based on previous work from cited papers because grid search was too computationally expensive.

**Prediction during inference with simultaneous truth and performance level estimation.** Simultaneous truth and performance level estimation (STAPLE) is an algorithm that was originally developed to estimate a single maximum likelihood segmentation from multiple independent segmentations of the same object [36]. For the 2.5D U-Net model, one prediction was made for each of the three orthogonal planes. Then, the STAPLE method was used during inference to obtain a single maximum likelihood segmentation from individual segmentations inferred from three orthogonal planes.

**3D nnU-Net.** The 3D nnU-Net model was trained and validated on the same data as the in-house 2.5D U-Net model. Patch sizes of (128,128,128) were used during training and inference. Foreground class oversampling, mirroring, zooming, Gaussian noise, Gaussian blur, brightness, and contrast were all used as augmentation techniques. The optimizer used was Adam, and the loss function was the average of DSC and cross-entropy losses.

The framework also used Automatic Mixed Precision (AMP) which greatly increased speed [37]. For testing, the model used test time augmentation (TTA), which has been shown to improve results [38].

The weights with the highest DSC score on the validation data were saved and, after finishing training, used for the test data evaluation. This highly optimized model was able to train for 600 iterations before it was forced to stop when the validation score did not increase after 100 iterations.

**Deep Bayesian (Hypermapp3r) comparison model.** We ran this comparison model exclusively on the test datasets and used 'out of the box' pre-trained weights. The model was trained using an augmentation scheme that consisted of four transformations: flipping along the horizontal axis; random rotation by an angle $\alpha \pm 90$ along the y and z axes; and the addition of Rician noise generated by applying the magnitude operation to images with added complex

noise, where each channel of the noise is independently sampled from a Gaussian distribution with random $\sigma = (0.001, 0.2)$ and changing image intensity by $\gamma = (0.1, 0.5)$.

Since this model requires a brain mask, the test data was segmented using the HD-BET segmentation tool [39]. We then inferred our data using the Deep Bayesian model. We used the batch command for their software "seg_wmh" with the paths for FLAIR-weighted images, T1-weighted images, and a brain mask. This model used two input channels—FLAIR-weighted and T1-weighted—compared to the in-house trained models, which only used one FLAIR-weighted image input. During inference, the model generated twenty predictions per participant to develop an uncertainty map of the segmentation. For this reason the full prediction time took longer than the other models.

### Test metrics

We used the following metrics on the test datasets, chosen to align with the results from the MICCAI WMH challenge:

- Dice similarity coefficient (DSC) [40].

- Hausdorff distance [41] (HD95, modified, 95th percentile in mm). For this metric, smaller values represent better performance.

- Average volume difference (AVD, in percentage). For this metric, smaller values represent better performance.

- Recall for individual lesions [42]

- F1-score for individual lesions [43]

The implementation of these metrics was found on the MICCAI WMH Challenge website.

The metrics used in some examples are true positive (TP) which is the overlap between annotated data and prediction. False positives (FP) are where there are predictions but no annotation, and false negatives (FN) are where there are no predictions but there is annotation.

### Test statistics

Since the performance metrics in the test dataset, in general, were not found to be normally distributed using the Shapiro-Wilk test, we used non-parametric test statistics. Friedman's related samples from Analysis of Variance (ANOVA) by Ranks were used to test the null hypothesis that the distribution of each test metric across subjects was the same for all three models, whereas the alternative hypothesis was that they were not the same. An additional pair-wise comparison was performed between the models. Statistical analyses were performed in SPSS V.28.0 (IBM SPSS Statistics). The significance level was set to $p = .05$.

## Results

In this section, we show the segmentation results of the models.

### Segmentation results

The results for the validation data for the two in-house trained models are shown in Fig 2.

Both models' performances were lesion size-dependent, with better performance for 3D nnU-Net in the larger lesions, as shown in Fig 3. The 2.5D model is shown to be more sensitive as the recall is generally higher, but with lower precision compared to the nnU-Net model.

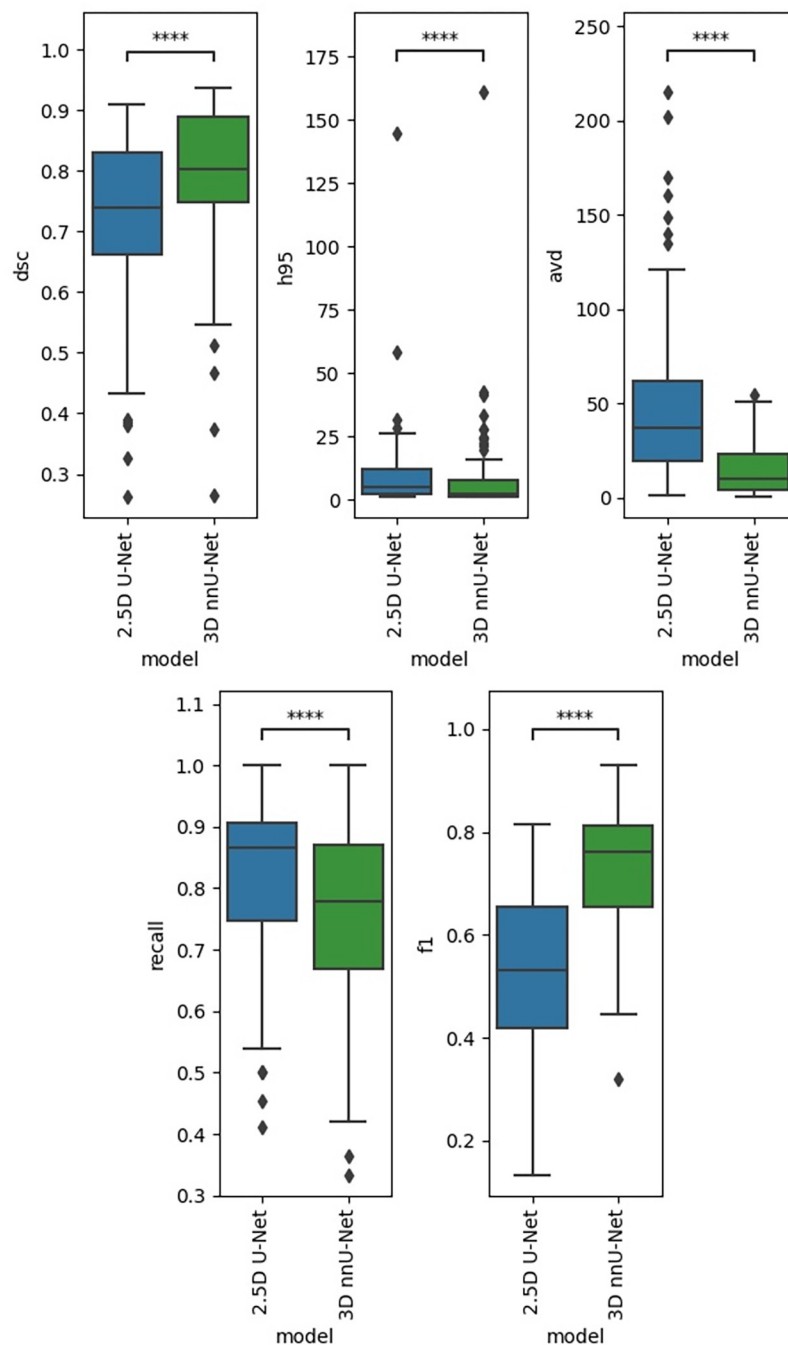

**Fig 2. Validation data results.** Performance results in the validation data. ns: p < = 1.00e+00, *: 1.00e-02 < p < = 5.00e-02, **: 1.00e-03 < p < = 1.00e-02, ***: 1.00e-04 < p < = 1.00e-03,****: p < = 1.00e-04.

**Test data segmentation results: All three models.**   The 3D nnU-Net model performed the best for all metrics except Recall on the internal dataset and h95 in the external dataset (see Figs 4 and 5).

For the internal test set, average (+/- std.dev) DSC scores were 0.68 (+/- 0.17), **0.76 (+/- 0.16)**, and 0.61 (+/- 0.23) for the 2.5D U-Net, 3D nnU-Net, and Deep Bayesian models,

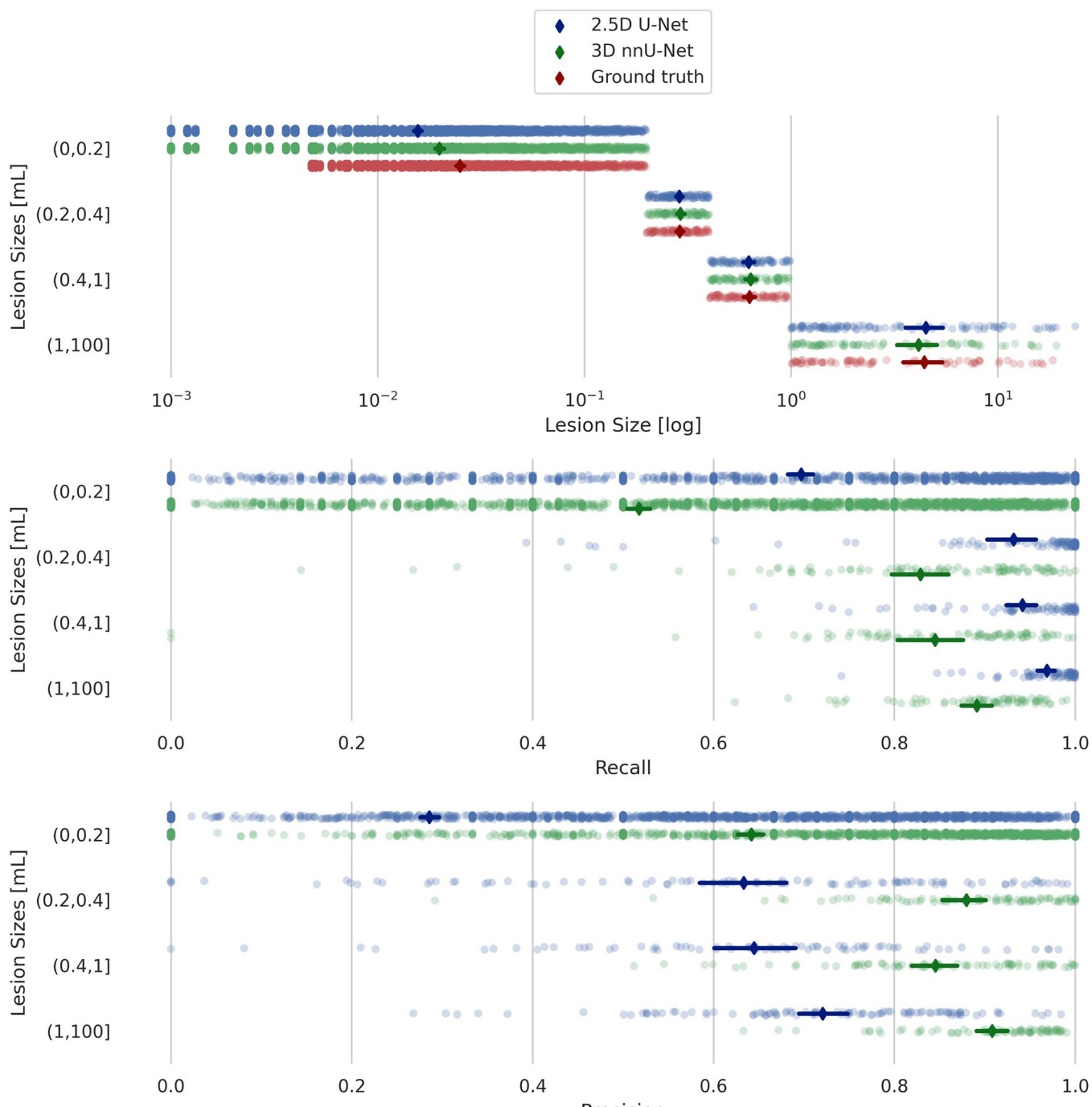

**Fig 3. Lesion size results from validation data.** Recall and precision scores for 2.5D U-Net versus 3D nnU-Net, grouped according to lesion size for the validation data.

respectively. The corresponding DSC scores for the external test set were 0.55 (+/- 0.20), **0.67 (+/- 0.16)**, and 0.54 (+/- 0.17).

Overall, there were significant differences in model performance for all metrics and both test sets (Related-Samples Friedman's Two-Way Analysis of Variance by Ranks, p < 0.001). The 3D nnU-Net was found to have the best performance for most of the metrics compared to

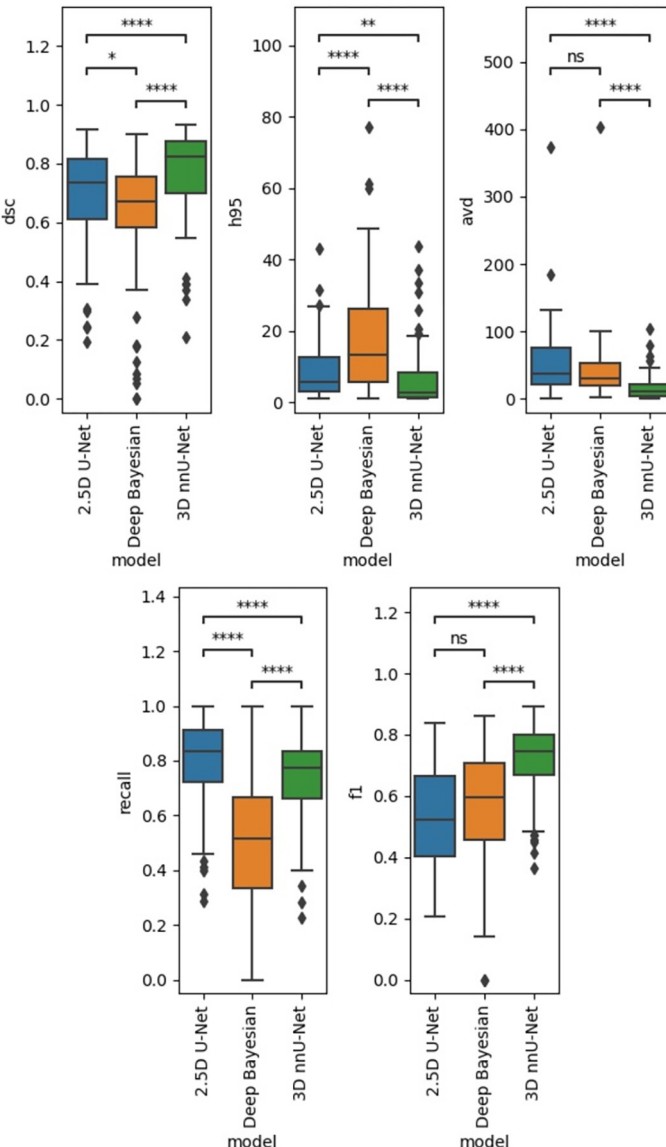

**Fig 4. Internal test data results.** Performance results in the internal test data. ns: p < = 1.00e+00, *: 1.00e-02 < p < = 5.00e-02, **: 1.00e-03 < p < = 1.00e-02, ***: 1.00e-04 < p < = 1.00e-03, ****: p < = 1.00e-04.

the 2.5D and Deep Bayesian models. Figs 6 and 7 show the recall and precision as functions of lesion size for the internal and external test datasets, respectively. As expected, the performances are trending upward as the lesion sizes increase.

Fig 8 shows sample cases of segmentation performance for all three models for three different participants in the internal test set. Here, the yellow color denotes over-segmentation, the red color denotes under-segmentation, and the green color denotes correct segmentation.

Large lesions appear to be detected with a high level of ground truth overlap. There seems to be more uncertainty, especially for the 2.5D and Deep Bayesian where the lesions are more smudged out.

Fig 9 shows the example segmentation from three different participants in the external test dataset for the three different models.

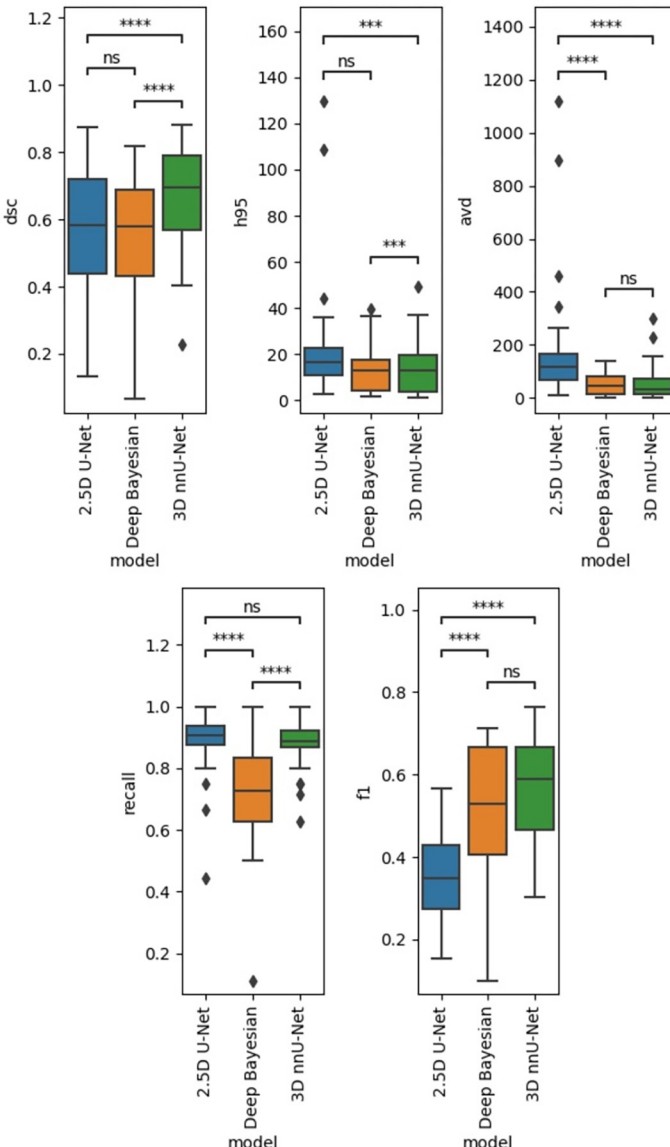

**Fig 5. External test data results.** Performance results in the external test dataset. ns: p < = 1.00e+00, *: 1.00e-02 < p < = 5.00e-02, **: 1.00e-03 < p < = 1.00e-02, ***: 1.00e-04 < p < = 1.00e-03,****: p < = 1.00e-04.

## Discussion

In the present study, we evaluated the performance of 2.5D and 3D nnU-Net CNN models for their ability to segment WMH based on volumetrically acquired FLAIR-weighted MRI sequences across a range of older adult participants and in the context of a multi-center pre-dementia imaging initiative. We compared these architectures against a recently published SOTA Deep Bayesian model. The present study is novel because it involved a large sample of participants from a national multi-center pre-dementia case-control study in which near-isotropic 3D FLAIR-weighted images were acquired and domain experts carefully reviewed segmentations to establish a reliable ground truth.

The ground truth as previously mentioned in the method section was developed by using an established method that can be rerun if the domain expert was not satisfied. This method is

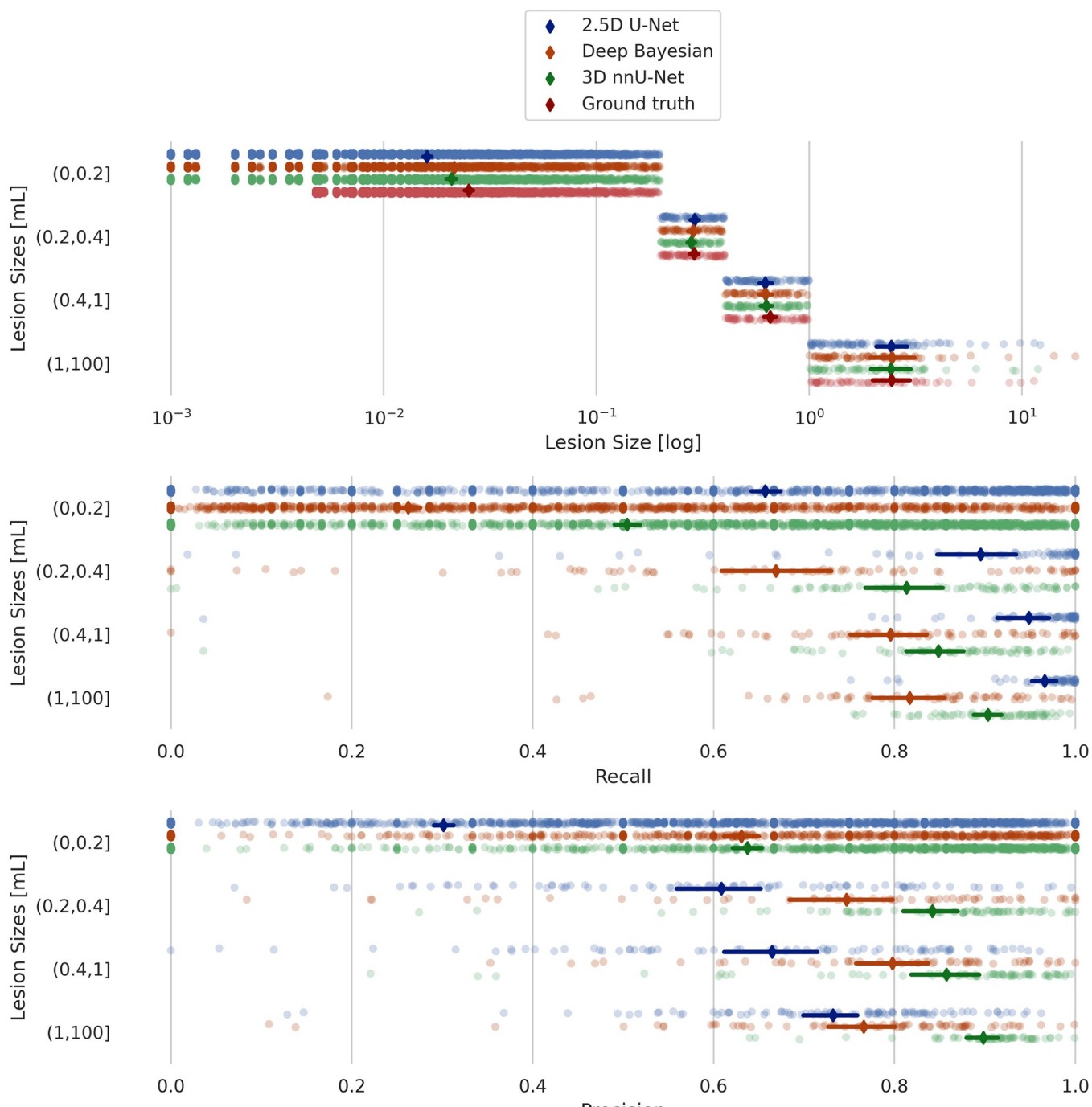

**Fig 6. Internal test data results.** Lesion recall and lesion precision scores for 2.5D U-Net, 3D nnU-Net and Deep Bayesian, grouped according to lesion size for the internal test data.

not as good as if the annotation were made by hand, but generating this many examples manually is not feasible in practice. In these experiments we avoided comparing to non deep-learning methods as our aim is to get a model in production with supervision and iterate upon the model with new data and update based on new feedback in a production setting. Most deep learning approaches seem promising as seen in the WMH challenge, hence we chose models

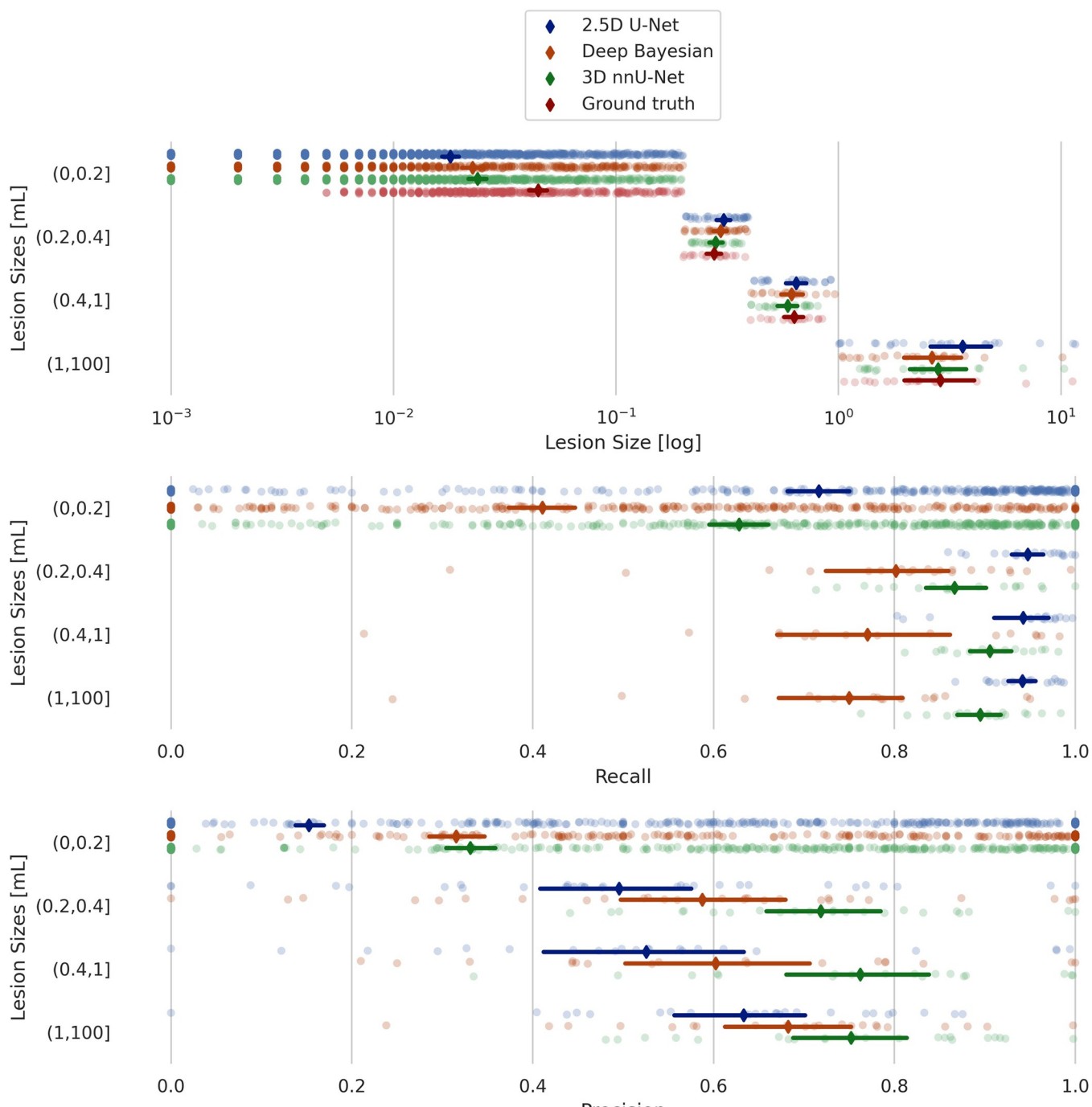

**Fig 7. External test data results.** Lesion recall and lesion scores for 2.5D U-Net, 3D nnU-Net and Deep Bayesian, grouped according to lesion size for the external test data.

for comparison that are somewhat unique. The 2.5D -model had promising results in the authors master thesis and covers an example from the 2D models while the Deep Bayesian model covers data from both out of sample T1 and FLAIR with interesting prediction outputs such as probability maps. The ability to get a good probability map for the prediction can be

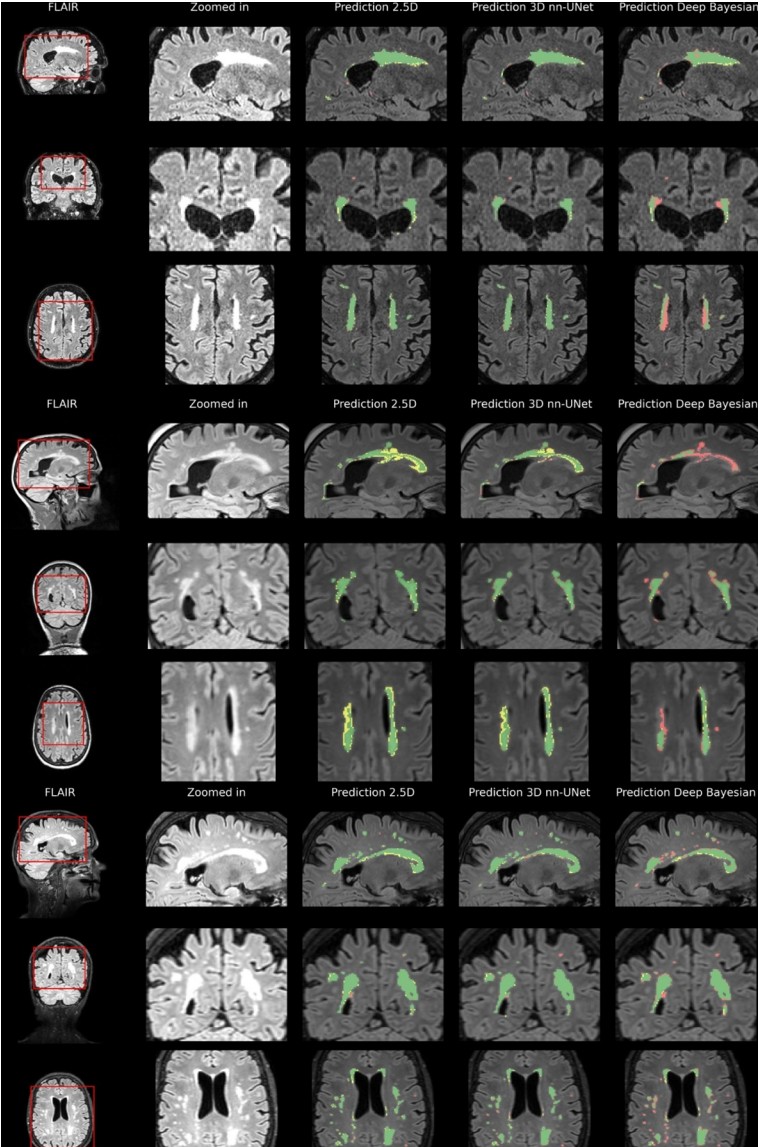

**Fig 8. Internal test data results.** Predictions based on internal test data overlaid on FLAIR for three participants. The first column shows three slices from each orientation for three different participants. The second column shows a zoomed-in view of an area of interest. The last columns show the segmentation results for the three different models. Green shows true positive voxels, red shows false negatives, and yellow shows false positives.

important for interpretation. This could be added to the 3D nnU-Net by either having an ensemble or adding MC dropout layers to the model.

In larger, previously reported research projects such as the MICCAI segmentation challenge, 2D FLAIR-weighted together with 3D T1 were used as data. The Deep Bayesian model used as an external test model in our research was trained on combined imaging features from 3D T1- and 2D FLAIR-weighted images by the original authors. This model was tested to see if the "out of the box" solution could be a better solution and to compare if adding T1 would improve the results dramatically.

The two in-house trained models in our study were exclusively using 3D FLAIR-weighted images. From a clinical perspective, it is appealing to produce an automated WMH

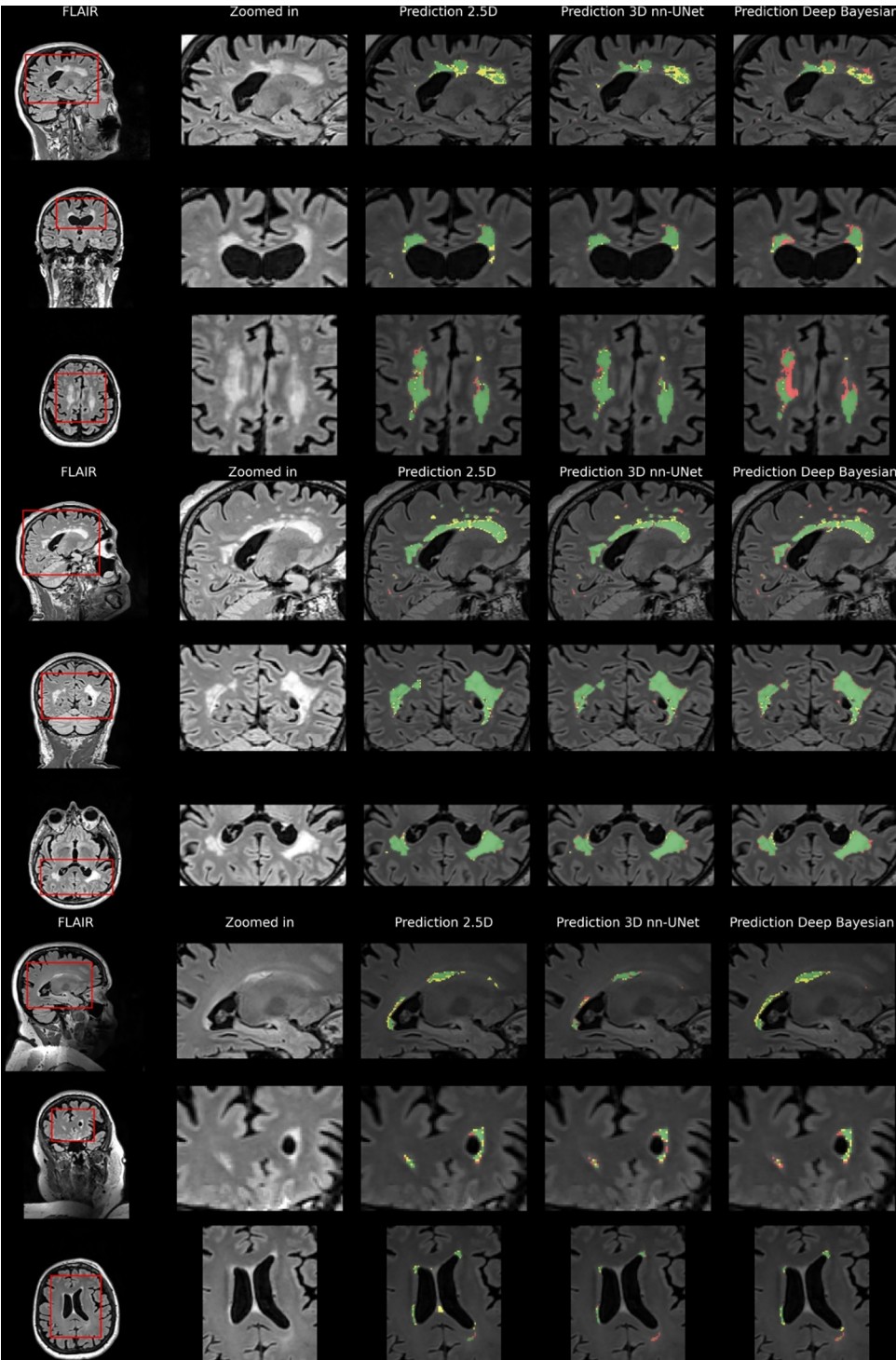

**Fig 9. External test data results.** Predictions based on external test data overlaid on FLAIR for three participants. The first column shows three slices from each orientation for three different participants. The second column shows a zoomed-in view of an area of interest. The last columns show the segmentation results for the three different models. Green shows true positive voxels, red shows false negatives, and yellow shows false positives.

segmentation using only one MRI sequence, obviating the need for new quality control and/or co-registration of multiple image inputs while also reducing complexity and barriers to adoption in a clinical setting.

The 2.5D and 3D U-Net models performed favorably in comparison with the reference SOTA Deep Bayesian model, if DSC score is concerned. Overall, the in-house-trained 3D nnU-Net model had the best performance across all of the test metrics. As expected, the performance of all three models was strongly dependent on lesion size, with a rapid drop in performance for individual lesion volumes of less than 0.2 mL.

The reason for the low recall and $F_1$ scores in the smallest lesions is the severe penalty for just misclassifying a few voxels. Detecting tiny lesions is of interest and potentially clinically meaningful. Further work will aim to increase the number of cases with small lesions across multiple scanner types. The 2.5D model seems to detect the smaller lesions better, with an average recall of ~0.63, at the cost of precision, at ~0.25 for the smallest lesions.

Interestingly, the reference model's performance was inferior for all metrics in our test data compared to the previously published test results from the Deep Bayesian model. A DSC of 0.61 (+/- 0.23), compared to the previously reported value of 0.893 (+/- 0.08). It must be declared that no new training was given to the reference model. However, these performance differences highlight the need for broad and diverse deep learning models [44]. It is also worth mentioning that the in-house trained models had in excess of 7 M parameters, whereas the Deep Bayesian model only has 515 K parameters (MC dropout). One might expect that models with more parameters should be able to learn more subtle image features, but this could also make them more susceptible to overfitting. Several sources could have contributed to differences in performances for the models, i.e. data inputs, image processing steps, and architecture. Therefore, it is important to note differences in performance on a relative but not an absolute scale.

It would be interesting to increase the number of parameters in the Deep Bayesian, given its good performance in its current form. The Deep Bayesian also generates an uncertainty map that can be very useful for further analysis. As previously stated, an ensemble of the in-house trained models could be attempted in the future to obtain similar uncertainty maps and reduce variance.

We also observed a ~12% drop in DSC on the external test data. Although the data sets were acquired with a similar 3D FLAIR-weighted protocol, there were some clear differences in the intra-scanner distributions, as shown in the first figure. The external test dataset only contained images from a single Prisma scanner, while the training data only included 11 cases from Prisma. The lack of training cases from this scanner type may explain the reduced model performance for the external dataset. Another possibility is that the distribution for the total WMH volume in the external test data had a median and IQR of 0.986 mL (0.611, 2.652) mL and a total range of (0.123, 21.252) mL, which was lower than the internal test data with a median and IQR of 3.35 mL (1.14, 5.0) mL and a total range of (0.15, 22.90) mL. Since the external dataset had a smaller median WMH volume, the lower scores could have been caused by the recall/precision being generally worse for the smaller lesions.

Although the intensity distributions for the different scanner types are mostly overlapping, we could apply an adaptive histogram normalization method as part of the pre-processing procedure [45] to make the distributions the same shape for all scanners or apply a brain extraction algorithm to remove non-brain tissue from the analysis [46]. This was not performed in our study as we wanted to avoid excessive pre-processing requirements.

## Limitations

First, although the in-house trained models were trained and validated on a large dataset (n = 375 participants across five national sites and six scanner types), the test sets of n = 66

(internal) and n = 29 (external) were modestly sized. Ultimately, these sample sizes reflect a degree of pragmatism to train, evaluate, and deploy models while recognizing the bottleneck of performing 470 ground truth segmentations.

Second, the in-house trained models were only trained on FLAIR-weighted data, whereas the reference model was trained on a combination of FLAIR-weighted and T1-weighted data. Our results do, however, suggest that T1-weighted images may not be needed for WMH segmentation. More explicit work on this is warranted. An issue with a direct comparison of the two in-house trained models is that they have different post-processing methods for predictions. The 2.5D model uses STAPLE on predictions from all three orientations, while the 3D nnU-Net uses test-time augmentation. The idea behind this is that we wanted to see what the best configurations from previous literature for the models would result in without the need for extensive parameter searches.

Third, the test and validation data sets were not annotated manually from scratch. Rather, annotations were based on segmentation using an established non-CNN-based algorithm, which was then reviewed and approved by one or two domain experts for the final ground truth. Given the large amount of data in the 3D series (> 200 2D slices per 3D FLAIR volume), fully manual annotation of the complete test dataset was not feasible in our current setup.

Finally, the models investigated have not yet been used prospectively, so it remains to be determined whether these CNN tools can be used across a wide range of patients. This will be the important final step in investigating clinical utility and whether this is a robust tool for widespread national use in ongoing national AD MRI studies.

There are certain considerations that have to be taken into account when comparing the in-house-trained models with the reference model. Making a direct comparison between the in-house and external models is not fair as they are not trained on the same cohort. The reason for including an external model is to check the difference between off-the-shelf trained models and in-house trained models in an effort to see the generalizability of SOTA-level models. The orientation of the training data can also affect the models in a way that, if it isn't considered, can lead to poor performance in testing. There is also a difference in the pre-processing and post-processing methods between the models.

## Conclusions

Using 3D FLAIR-weighted images acquired at both 1.5T and 3T, automated and reliable WMH segmentation was achieved. Our findings indicate that models relying on single 3D-acquired FLAIR image inputs produced WMH segmentation masks that were comparable and slightly better than a SOTA approach that used pairs of MR images. This research helps to support efforts towards large scale WMH assessments, particularly for our national Norwegian clinical research initiative. The 3D nnU-Net had the highest performance scores; thus, we will continue to test it on new data and automate the segmentation of WMH in our ongoing AD-related research projects.

## Author Contributions

**Conceptualization:** Martin Soria Røvang.

**Data curation:** Martin Soria Røvang, Per Selnes.

**Formal analysis:** Martin Soria Røvang, Carole Sudre.

**Funding acquisition:** Tormod Fladby.

**Investigation:** Martin Soria Røvang.

**Methodology:** Martin Soria Røvang, Tormod Fladby.

**Project administration:** Martin Soria Røvang, Tormod Fladby.

**Software:** Martin Soria Røvang, Carole Sudre.

**Supervision:** Martin Soria Røvang.

**Validation:** Martin Soria Røvang.

**Visualization:** Martin Soria Røvang.

**Writing – review & editing:** Per Selnes, Bradley J. MacIntosh, Inge Rasmus Groote, Lene Pålhaugen, Carole Sudre, Tormod Fladby, Atle Bjørnerud.

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
