## [Decision Letter · Decision Letter 0]

6 Oct 2022

PONE-D-22-24185Segmenting white matter hyperintensities on isotropic three-dimensional Fluid Attenuated Inversion Recovery magnetic resonance images: A comparison of Deep learning tools on a Norwegian national imaging databasePLOS ONE

Dear Dr. Røvang,

Thank you for submitting your manuscript to PLOS ONE. After careful consideration, we feel that it has merit but does not fully meet PLOS ONE’s publication criteria as it currently stands. Therefore, we invite you to submit a revised version of the manuscript that addresses the points raised during the review process.

Ultimately the two Reviewers had conflicting recommendations regarding the decision. Rather than recruiting a third Reviewer, I would like you to carefully address the points raised by both Reviewers.  In addition, please address the following point: - Avoid the use of the word "significant" unless in a statistical context (e.g., line 83)

We look forward to receiving your revised manuscript.

Kind regards,

Niels Bergsland

Academic Editor

PLOS ONE

Journal Requirements:

Reviewers' comments:

Reviewer's Responses to Questions

**Comments to the Author**

1. Is the manuscript technically sound, and do the data support the conclusions?

Reviewer #1: Yes

Reviewer #2: No

2. Has the statistical analysis been performed appropriately and rigorously? 

Reviewer #1: Yes

Reviewer #2: No

3. Have the authors made all data underlying the findings in their manuscript fully available?

Reviewer #1: Yes

Reviewer #2: No

4. Is the manuscript presented in an intelligible fashion and written in standard English?

Reviewer #1: Yes

Reviewer #2: No

5. Review Comments to the Author

Reviewer #1: The authors investigated the use of deep learning approaches to segment 3D FLAIR data. They collected and used data from a large multisite dementia initiative to test two 3D deep models for segmenting WMH (one trained internally and one without any further training) compared to a Bayesian deep network. The models were trained on a subset of 642 participants and tested on a subset as well as an external dataset. Some subset of data was manually segmented to evaluate and train the model. They found that the 3D nnU-net performed best outperforming other models with some caveats. This is an interesting analysis, however there are some issues that should be addressed:

1. In the 2.6.1 section, many of these decisions are described but not critically reasoned. It would be helpful to justify the reasoning behind various parameter choices in the manuscript (even if they are just based on prior experience and work).

2. The performance on the external dataset is quite low in general. The authors should comment on whether intensity normalization approaches may help generalize these types of approaches.

3. The authors state: “However, since the models tested here were not trained on the MICCAI dataset, a direct comparison of performance may be misleading.” The MICCAI is an open challenge, did the authors consider submitting results to identify how well they performed in that dataset? Otherwise, if this is not important then these types of statements should be removed.

4. The authors state: “Our results do, however, suggest that T1-weighted images may not be needed for WMH segmentation.” Why is this important especially given that a T1-weighted image is almost always available if a FLAIR is available?

5. The conclusions of the manuscript leave a lot to be desired. While the initial motivation for the paper seems relevant, the conclusions are quite flat. For instance, they state that models can be trained on 3D imaging data, it is unclear why this is important to establish. This seems like something that should be feasible. They also state that a T1-weighted image would not be necessary, but this (see previous comment) is also not that important given that they are often available (unless there is motion).

6. The authors also should revisit their hypotheses from the introduction explicitly.

7. It is unclear how many manual segmentations were completed. How many slices were manually segmented?

8. In the introduction, the authors should include a paragraph on why it is important to develop 3D FLAIR segmentation algorithms. What challenges exist in this space? What has been done previously? Are there any tools that have been previously developed for 3D FLAIR? If so, why are they not used here? Or why do they not work?

Minor

1. The figure resolutions on all images are quite poor. They should all be redone to higher resolution images.

2. Tables 3 and 5 show the same results as figures 2 and 3. Just one set can be presented.

3. Some results are presented multiple times in the text, tables, and figures. For instance, for the internal test – the authors state: “For the internal test set, average (+/- std.dev) DSC scores were 0.70 (+/- 0.13), 0.78 (+/-0.10), and 0.63 (+/- 0.15), for the 2.5D U-Net, 3D nnU-Net and Deep Bayesian models, respectively.” They then present this in table 5 and figure 3. It would be better to show it, for instance, in one table and then just state the finding: “3D nnU-Net performed best across all metrics in the internal data (see table 5).”

Reviewer #2: The authors present an evaluation of 3 models/algorithms for segmenting white matter hyper-intensities (WMH) on 3D T2 FLAIR MRI images. One of the models has been implemented and trained by the authors. The second model is the default nnUnet implementation, and the third one is a deep Bayesian model. Two of these models are trained on the Norwegian Disease Dementia Initiation (DDI) dataset, and tested on a subset of the data associated with this study, as well as on an additional external dataset. The main difference between these two networks and the Bayesian one is that the later was trained on combinations of T2-FLAIR and T1 acquisitions.

The main outcomes are: a demonstration that segmenting WMH on 3D T2-FLAIR acquisitions is feasible, and that the nnUnet gives the best evaluation metrics.

English language skills are globally satisfactory, but the paper structure is sometimes difficult to follow. As an example: there is a pre-processing section, 2.5, but data pre-processing (related with the nnUnet) is not detailled here, rather in a later section, 2.6.2. Inconsistencies across section 2.7 and section 2.8 and the results create additional confusion. While test metrics and statistics are presented in section 2.7, presentation of results start with a not introduced section about WMH volume distribution of the data. In addition, TP, FN and FP are used as a performance indicators, but they are not presented in sections 2.7 and 2.8.

In the introduction, the authors assess that so far WMH segmentations have mostly been attempted on 2D FLAIR acquisitions. Could you please provide citations (line 102, line 111 again) ? Nevertheless, one cited paper (Forooshani et al., 2022), is not discussing the use of 2D images. Instead, a 3D Bayesian convolutional neural network is proposed. A confusion between T2-FLAIR and 2D FLAIR may have occurred ?

Moreover, the two articles presented as state-of-the art solutions are citation 5 (citation number 7) and citation 6 (citation number 10): perhaps there is more out there in the literature ? In addition, the Bayesian model cited as SOTA solution requires the use of both T1 and T2 acquisitions, while the objective of the article is to perform segmentation on T2-FLAIR, which creates some confusion.

Some parameters in the equations, as in equation (1), are not explained by the authors (alpha and beta in this case).

The level of details is unbalanced: on one side there is an extensive description of Python libraries for splitting data into train/val subsets, on the other the N4 implementation or DICOM conversion tools are not detailed at all. The latter appears to me as of higher value for reproducibility purpose. Also, details are provided concerning the DDI study, that are of interest, but not of the highest value regarding the work here presented. The reader is being referred to another article to get this valuable information.

Inconsistencies arise between text and tables. Regarding the DDI study, 5 national sites are mentioned, whereas 13 institutions are presented in the corresponding table. The origin or meaning of “institutions” should be explained, as it is difficult for the reader to associate them with the written details. Lines 387-389: three intervals are associated with 4 values, another example of inconsistency.

While the fist model is presented as the in-house model at the beginning of the article, the nnUnet implementation is also referred as an in-house solution in later sections. This creates confusion: the nnUnet is not a solution designed and implemented by the authors (see line 410 just as an example).

The methods are too different to be compared. In fact, they differ in terms of input needs (1 vs 2 contrasts), pre-processing, training and validation splitting (what would have other folds of the nnUnet cross-validation procedure generated as results ?) and prediction (test time augmentation in the case of nnUnet). This level of differences makes it hard to appreciate the cause of different performances. Would some technical choices made in the nnUnet model help the in-house model to learn better on the training set ? I suppose that having at least the same pre-processing and prediction steps would be necessary for sake of comparison.

An additional concern is associated with the use of the results obtained during the MICCAI WMH segmentation challenge as scores to which the proposed algorithms shall be compared. Nevertheless, comparing results obtained on different data in a single table appears as unsuitable to establish relevant conclusions.

Lines 423-428 are unclear. Is the objective of the work to provide a clinically relevant solution, or to obtain the best performance scores ?

In conclusion, the fact that nnUnet performs the segmentation of WMH on 3D T2-FLAIR acquisitions is an interesting result. The comparison with the two other models does not appear relevant because of the level of differences in implementation, inputs requirements and results. The aspect concerning work performed on 2D T2-FLAIR acquisition should be clarified, as 3D acquisitions are a standard.

6. PLOS authors have the option to publish the peer review history of their article (what does this mean?). If published, this will include your full peer review and any attached files.

Reviewer #1: No

Reviewer #2: No

---

## [Author Response · Author response to Decision Letter 0]

2 Dec 2022

Response has been added as an attachment

---

## [Decision Letter · Decision Letter 1]

26 Jan 2023

PONE-D-22-24185R1Segmenting White Matter Hyperintensities on Isotropic three-dimensional Fluid Attenuated Inversion Recovery Magnetic Resonance Images: Assessing Deep Learning Tools on a Norwegian Imaging DatabasePLOS ONE

Dear Dr. Røvang,

Thank you for submitting your manuscript to PLOS ONE. After careful consideration, we feel that it has merit but does not fully meet PLOS ONE’s publication criteria as it currently stands. Therefore, we invite you to submit a revised version of the manuscript that addresses the points raised during the review process.

As was the case with the initial submission, the two Reviewers have again provided me with conflicting recommendations. It appears to me that there is still a considerable amount of work that can be done to improve your manuscript. Please carefully consider each of the concerns raised by Reviewer 2.

We look forward to receiving your revised manuscript.

Kind regards,

Niels Bergsland

Academic Editor

PLOS ONE

Reviewers' comments:

Reviewer's Responses to Questions

**Comments to the Author**

1. If the authors have adequately addressed your comments raised in a previous round of review and you feel that this manuscript is now acceptable for publication, you may indicate that here to bypass the “Comments to the Author” section, enter your conflict of interest statement in the “Confidential to Editor” section, and submit your "Accept" recommendation.

Reviewer #1: All comments have been addressed

Reviewer #2: (No Response)

2. Is the manuscript technically sound, and do the data support the conclusions?

Reviewer #1: Yes

Reviewer #2: Partly

3. Has the statistical analysis been performed appropriately and rigorously? 

Reviewer #1: Yes

Reviewer #2: Yes

4. Have the authors made all data underlying the findings in their manuscript fully available?

Reviewer #1: No

Reviewer #2: No

5. Is the manuscript presented in an intelligible fashion and written in standard English?

Reviewer #1: Yes

Reviewer #2: Yes

6. Review Comments to the Author

Reviewer #1: (No Response)

Reviewer #2: I would like to recognize and thank the authors regarding the work performed in order to address some of the concerns raised during this first exchange. Notably, precision concerning the different kinds of acquisitions, what was performed in-house and not, the concision effort on some parts of the article and the development of aspects that required additional explanations are very welcome.

Nevertheless, despite these positive improvements, this new version raises a lot of new interrogations, that I would consider at least as important as the ones discussed during the first round.

The objective of the article, from my understanding, is to report work performed in order to provide an automated WMH segmentation from 3D FLAIR images to clinicians. Work has been done and is available in the literature on this topic, but presents differences with the one reported here. Deep learning-based methods handling this segmentation task have been proposed, but based on 2 images, 3D-T1 (2D T1 is still mentioned instead of 3D in the introduction) and 2D FLAIR. Other methods use 3D FLAIR, but are not based on deep learning methods.

I disagree with the authors regarding their response to reviewer 1 (R1.8), as the main objective of this work being to segment WMH from 3D FLAIR data, the other methods related to this application should be compared to the proposed models. Indeed, despite the fact that DL-based methods have been shown to provide increased performance metrics on a number of applications compared to previous works, the “superiority” of the DL-based models needs to be proven against gaussian-mixture models (GMM) or the method proposed in Zhong et al, 2014 for this particular case. Moreover, the authors mention the use of GMM as part of their semi-automated annotation procedure, which I understand as the authors having effectively used this method, which would have then been easily compared. This comparison would have been, in my opinion, at least as meaningful as the comparison with a 2 input images model, from which we could have expected the lower performance metrics, due to the absence of this bayesian model retraining.

The medical images processing community is very familiar with the Unet architecture and the nnUNet framework, but not with the models presented here as state of the art (Hypermapp3r) or the 2.5-D Unet. In particular, the selection of the 2.5-D model should be more motivated, and the choices made to generate it really detailed in the publication, as the reference provided to the readers for explanations is the main author’s master thesis report.

In the results and discussion sections, the nnUNet model is presented as the one providing the overall best performance metrics. From the clinical point of view, is the coverage of the large lesions of the highest importance, or the capability of a model to actually detect lesions, even the smaller ones ? If detecting lesions has a significant meaning for clinicians, is the selection of the nnUNet model still relevant ? For an example of lesions detection evaluation, an interesting reference could be Commowick et al., 2018 (doi: 10.1038/s41598-018-31911-7)

The Hypermapp3r model is presented as having some appreciable advantages. The lower number of parameters, compared to the 2 other models, would be an advantage regarding model’s deployment, and this bayesian model can provide uncertainty related information. But the perspective of retraining / fine-tuning this model or adapting a model with these properties to the 3D-FLAIR WMH application is not envisaged.

The authors did not provide information about the reason of the population description changes (number of subjects modified from 642 to 441, that influences all the related subsets).

Overall, despite interesting aspects presented in this paper, I understand it as a compilation of valuable work, that unfortunately lacks investigations and does not allow to construct and respond to the story that could be attached to the main objective.

7. PLOS authors have the option to publish the peer review history of their article (what does this mean?). If published, this will include your full peer review and any attached files.

Reviewer #1: No

Reviewer #2: No

---

## [Editor Report · Decision Letter 2]

28 Apr 2023

Segmenting White Matter Hyperintensities on Isotropic three-dimensional Fluid Attenuated Inversion Recovery Magnetic Resonance Images: Assessing Deep Learning Tools on a Norwegian Imaging Database

PONE-D-22-24185R2

Dear Dr. Røvang,

We’re pleased to inform you that your manuscript has been judged scientifically suitable for publication and will be formally accepted for publication once it meets all outstanding technical requirements.

Kind regards,

Niels Bergsland

Academic Editor

PLOS ONE
---

## [Editor Report · Acceptance letter]

15 Aug 2023

PONE-D-22-24185R2 

Segmenting White Matter Hyperintensities on Isotropic three-dimensional Fluid Attenuated Inversion Recovery Magnetic Resonance Images: Assessing Deep Learning Tools on a Norwegian Imaging Database 

Dear Dr. Røvang:

I'm pleased to inform you that your manuscript has been deemed suitable for publication in PLOS ONE. Congratulations! Your manuscript is now with our production department. 

Kind regards, 

on behalf of

Dr. Niels Bergsland 

Academic Editor

PLOS ONE